# Investment Opportunities for mRNA Technology in Low- and Middle-Income Countries: Key Findings and Future Perspectives

**DOI:** 10.3390/vaccines13020112

**Published:** 2025-01-23

**Authors:** Ariane de Jesus Lopes de Abreu, Cheleka A. M. Mpande, Matthias Helble, Martin W. Nicholson, María de los Ángeles Cortés, María Eugenia Pérez Ponsa, Ivan Redini Blumenthal, Francisco Caccavo, Tomas Pippo, Judit Rius Sanjuan, Claudia Nannei

**Affiliations:** 1The World Health Organization, Avenue Appia 20, 1211 Geneva, Switzerland; mpandec@who.int (C.A.M.M.); helblem@who.int (M.H.); nicholsonm@who.int (M.W.N.); nanneic@who.int (C.N.); 2The Pan American Health Organization, 525 23rd St NW, Washington, DC 20037, USA; cortesmar@paho.org (M.d.l.Á.C.); perezmar3@paho.org (M.E.P.P.); rediniiva@paho.org (I.R.B.); caccavof@paho.org (F.C.); pippoto@paho.org (T.P.); riussajud@paho.org (J.R.S.)

**Keywords:** LMIC vaccine manufactures, technology transfer, vaccines for epidemic and pandemic prevention, public investments and subsidies, vaccine access

## Abstract

In April 2024, a hybrid meeting organized by the WHO, PAHO, and MPP during the World Bank Spring Meetings focused on financing mRNA-based technologies in Low- and Middle-Income Countries (LMICs). This meeting sought to engage multilateral development banks (MDBs) and stakeholders in financing the expansion of vaccine production and enhancing pandemic preparedness. The COVID-19 pandemic underscored the disparities in vaccine production and distribution, highlighting the need for localized production to improve global health equity. The WHO’s mRNA Technology Transfer Programme, initiated in 2021, aims to build local capacity for mRNA vaccine development and manufacturing. Key sessions covered during the meeting include innovative investment models, with MDBs discussing funding instruments and the necessity of an integrated ecosystem for sustainable vaccine manufacturing. Challenges such as technological risks and the need for higher risk appetite were addressed, along with innovative financing mechanisms like blended financing. An analysis of capital and operational expenditures for mRNA vaccine facilities was presented, projecting significant production capacity in LMICs within a decade. Panelists emphasized the need for sustainable R&D investment and shared experiences in securing funding for mRNA technology. The meeting underscored the importance of collaboration, innovative financing, ecosystem development, and public–private partnerships, marking a pivotal step towards advancing mRNA technology in LMICs to tackle global health challenges.

## 1. Introduction

Vaccines are crucial in saving millions of lives every year, making them a vital part of the world’s public health success stories. They are cost-effective, improve life expectancy, and contribute to economic growth [1]. However, the progress in vaccinations varies greatly across countries, particularly in low- and middle-income countries (LMICs), where vaccination rates often remain low [2].

However, progress in vaccination coverage varies significantly across countries, especially in low- and middle-income countries (LMICs), where vaccination rates are often hindered by vaccine inequities driven by affordability, availability, and weakened health systems. Limited geo-diversification of vaccine production contributed to the vaccine inequity experienced during the COVID-19 pandemic, as countries where vaccines are produced decided to first fulfill their own markets over ensuring global equity. Additionally, other factors, such as research and development (R&D), production, supply, acceptance, and the administration of vaccine products worldwide, also affect disparities in vaccine access [3]. Consequently, the new emphasis on local (and regional) production demonstrates the need for interested countries to also strengthen R&D, innovation, and capacity-building to achieve sustainable vaccine production during inter- and intra-pandemic times. Proper ecosystem development is essential, and financing plays a crucial role.

Vaccine research, development, and manufacturing require significant capital, ranging from tens to hundreds of millions of dollars. While the financial commitment has to be substantial, the potential returns are equally significant. Sustainable vaccine manufacturing requires the continuous development, production, and supply of safe, effective, and affordable vaccines globally. To achieve this, pharmaceutical companies must have viable business models and incentives to manufacture vaccine products. This presents a unique opportunity for stakeholders to invest in a sector that not only promises financial gains but also contributes to regional and global health security [4,5].

The potential of the messenger ribonucleic acid (mRNA) technology for rapid scale-up and production of safe and effective vaccines was demonstrated during the COVID-19 emergency response, making it a promising tool to ensure equitable and global access to vaccine manufacturing and pandemic response [6].

The mRNA Technology Transfer Programme (mRNA TT Programme) was launched in 2021 as a response to the global COVID-19 vaccine access inequity. It is a collaboration between the World Health Organization (WHO), the Medicines Patent Pool (MPP), and private and public program partners from 15 LMICs in Africa, Asia, Europe, and South America (Figure 1). It aims to enable autonomy in the development, manufacturing, and distribution of mRNA vaccines in LMICs. One of the program’s expected results is to contribute to the overall improvement of health and health security in LMICs through equitable access to regionally produced mRNA-based vaccines [7,8].

The mRNA TT Programme differentiates from traditional models of technology transfer (unidirectional) by operating in a global collaborative network driven by a multilateral technology transfer approach, empowering its partners in LMICs to gain know-how and absorb technology (Figure 2).

The unwillingness to share the know-how of established mRNA technology required program partners to establish de novo mRNA technology, thus fostering a predominantly South–South mRNA R&D collaboration. The program aims to fulfill two complementary general objectives, to (a) establish and/or enhance sustainable mRNA vaccine manufacturing capacity; and (b) develop skilled human capital in the regions where mRNA vaccine manufacturing capacity is established and/or enhanced [7,8]. In the Americas, the PAHO’s Innovation and Regional Production Platform [9] contributes to the program’s implementation while undertaking complementary activities to strengthen regional ecosystems.

The mRNA technology transfer program has made substantive progress in building up a network of partners in 15 countries to acquire mRNA technology to produce vaccines. To enable the sustainability of investments and foster the growth of the mRNA vaccine industry beyond COVID-19, R&D consortia have been established to advance the application of the technology and develop vaccines for dengue, hand, foot and mouth disease (caused by enterovirus-A 71 (EV-A71)), malaria (caused by *Plasmodium vivax*), therapeutic human papillomavirus, pandemic and seasonal influenza, and leishmaniasis. Additionally, a global lipid consortium was developed to reduce the cost-of-good, freedom to operate, and to improve the thermostability of mRNA-based products. Implementing multilateral technology transfer models requires access to novel financial models that can support the development and manufacturing needs of emergent product development partnerships and manufacturers. Given that mRNA has extensive market potential across various disease areas beyond infectious diseases, the WHO has convened convening sessions for multilateral development banks (MDBs) to be introduced to the program partners and raise awareness of upcoming investment opportunities [10].

On 17 April 2024, a hybrid meeting was held during the World Bank Spring Meetings by the WHO, Pan American Health Organization (PAHO), and MPP. The meeting’s purpose was to discuss the potential for MDBs to invest in mRNA-based technologies in LMICs. This investment can help improve pandemic preparedness and response and address unmet public health needs.

This article aims to describe the meeting discussions related to challenges associated with financing mRNA vaccine manufacturing, the potential financial opportunities and mechanisms that can enable multilateral partnership for the development and manufacturing of innovative products, and to list the findings in relation to financial opportunities for mRNA vaccine R&D considering pandemic preparedness, prevention, and response (PPPR).

## 2. Meeting Design

The meeting was held on 17 April 2024, in Washington D.C. at PAHO headquarters. This date was selected to take advantage of the World Bank Spring meeting, taking place from 19 to 21 April 2024 in Washington, DC, ensuring the presence of a wide variety of representatives from international financial institutions. The list of participants included team members (from the WHO, MPP, and PAHO) of the mRNA TT Programme, the program partners, and representatives from MDBs to discuss the financial support required for expanding vaccine pipelines and pandemic capacity (Appendix A).

The objective of the meeting was to address the sustainability of pandemic response during inter-pandemic periods, to address local and/or regional unmet health needs in LMICs through the mRNA technology, and to highlight the need to financially support these activities to ensure expansion of the vaccine pipelines and pandemic capacity. As the program is rooted in multilateral product development and manufacturing partnerships, access to financing that can support this and also factor the emergent industry partners exist in is critical for the success of the program.

The meeting format was designed as a mix of presentations and panel discussions. Its sessions were focused on the importance of transferring platform technology and showcased multilateral development banks’ instruments for funding the research, development, and manufacturing of health products in LMICs.

The guiding topics for the sessions were divided as follows:Investment models of multilateral development banks for local manufacturing of health products;Reflections on capital expenditure and operating expense (CAPEX-OPEX) models and estimated capacity to contribute to a pandemic response within the context of the mRNA Technology Transfer Programme;Building the ecosystem for development of new mRNA-based products in LMICs;Addressing the needs of a nascent mRNA industry in LMICs;Envisaging new financing models for LMIC-based manufacturers (Appendix A).

## 3. Main Findings of Meeting Sessions

### 3.1. Session 1: Investment Models of Multilateral Development Banks for Local Manufacturing of Health Products

During this session, the focus was on presenting funding instruments for the research, development, and manufacturing of health products in LMICs. A panel of representatives from MDBs discussed their current approach and outlined the existing barriers and needs they face when considering new investments in health product development projects in LMICs.

The discussions considered the dynamics of the mRNA TT Programme as the starting point. Panelists recognized that technology transfer could not occur in isolation but requires an ecosystem that considers elements such as R&D and manufacturing within these ecosystems. Establishing manufacturing capacity takes time, as does bringing a product to commercial launch, underscoring the urgency of initiating these processes now for future pandemic preparedness.

The panel highlighted the need for inter-pandemic sustainability of manufacturing facilities and ecosystems, maintaining readiness for future pandemics through measures like establishing product pipelines and forming R&D consortia. The discussion explored various instruments for investing in R&D and manufacturing health products, aiming to foster innovation, improve access to essential medicines, and ensure long-term sustainability in global health preparedness.

Creating bankable projects within MDBs was emphasized as critical, not just for immediate gains but for fostering sustainable, regionalized health manufacturing ecosystems. This includes investing in infrastructure, human capital, regulatory resources, and procurement and financing capabilities. Despite MDBs’ long history of supporting local manufacturers in developing countries, they face significant challenges, especially regarding technology risks. The panel called for better collaboration among MDBs to pool resources and increase their risk appetites.

Innovative financing models were advocated, including time-bound market-shaping initiatives and milestone payments to incentivize commercial viability. The panel stressed the importance of reducing dependency on specific markets to achieve self-sustained ecosystems, democratizing innovation, and fostering regional incubation tailored to the needs of developing countries.

The Asian Infrastructure Investment Bank (AIIB) acknowledged its newcomer status in the health sector and its ongoing health strategy preparation. They noted the diverse demands for local pharmaceutical manufacturing, particularly post-pandemic, and the challenges posed by clients in weaker financial positions. AIIB underscored the need for a higher risk appetite and blended finance to support dynamic and innovative players in the health sector.

The Inter-American Development Bank (IDB) emphasized collaboration to create impactful value and preserve lives, especially in highly affected regions. They highlighted four key aspects for project analysis: sponsor experience and credibility, market risk mitigation, construction risk, and sustainability. The IDB is willing to fund early-stage R&D with grants and later-stage clinical developments with equity, stressing the need for clear exit strategies and long-term partnerships.

The International Finance Corporation (IFC) expanded its scope post-pandemic to include various stakeholders, developing strategies and partnerships to support inclusiveness in the health sector. The IFC emphasized the importance of derisking models and blended finance to catalyze more financing, especially in Africa, proposing innovative models like using public financing for vaccine manufacturing with repayment upon success.

The session concluded with discussions on financing expectations, risk management, and collaboration mechanisms among MDBs, private financiers, and public sector funders. The panelists highlighted the importance of patient capital, flexible financing structures, and collaborative efforts to support pharmaceutical manufacturing. An integrated approach for sustainable end-to-end biomanufacturing (Figure 3) involving early engagement and strategic alignment was deemed essential for project viability and sustainability, ensuring comprehensive support across the pharmaceutical manufacturing value chain in developing countries.

### 3.2. Session 2: Reflections on Capital Expenditure and Operating Expense (CAPEX-OPEX) Models and Estimated Capacity to Contribute to a Pandemic Response Within the Context of the mRNA Technology Transfer Programme

This session presented the results of capital expenditure (CapEx) and operating expense (OpEx) analysis conducted to assess the costs of establishing and maintaining mRNA production capability [11], followed by recent work examining the program’s potential contribution to mRNA vaccine availability in pandemic scenarios. Capital expenditure costs relate to investments in fixed assets, in this case the outlay required to set up and equip the manufacturing facility, whereas operating expense covers the ongoing costs required to run the facility.

The CapEx and OpEx analysis evaluated the investment requirements associated with setting up and operating small-scale good-manufacturing-practice-grade mRNA vaccine facilities, considering technical requirements and production capacities along with sustainability and long-term capacity retention. The presentation emphasized that decision-making regarding facility design and specifications lies with the program partners, with the analysis providing essential information to facilitate these decisions.

The analysis focused on balancing costs and capabilities, considering a small-footprint facility focusing on the key steps in mRNA vaccine production, including the transcription of mRNA itself and encapsulation to create the bulk mRNA vaccine drug substance. Other steps such as DNA template generation and fill and finish capability are excluded as they can potentially be outsourced to third parties.

Three scenarios were modeled: a standalone mRNA facility, an mRNA facility integrated into an existing vaccine manufacturer, and a flexible facility incorporating biologics manufacturing. The costs ranged from $15 to $20 million for standalone facilities, $5 to $10 million for integrated facilities, and $80 to $112 million for flexible facilities. Operational expenses varied across the scenarios, with annual mRNA production of 200,000 doses for each, with a maximum capacity of 15,000,000 doses on a single-shift basis, which is scalable as the number of shifts in a facility can be increased.

The subsequent part of the presentation discussed the mRNA technology transfer program’s potential contribution to mRNA vaccine availability during a pandemic situation and considered a future pandemic declaration at three-time points: 3, 5, and 10 years from now.

The analysis projected cumulative mRNA vaccine production from the program partners as a whole and assessed potential capacity across three basic scenarios: low-, mid-, and high-output cases. The model indicated that the cumulative, collective output from program partners ranged from 350 million to 1.2 billion doses over 18 months post-pandemic declaration, and depends on various factors such as partner capabilities, production systems, and underlying growth in mRNA vaccine requirements and usage. Upside factors such as an increase in the number of program partners and second-generation automated manufacturing technologies could enhance production capacity above the predicted range.

The conclusion highlighted the importance of establishing small-footprint facilities to balance investment costs with production capacity, maintaining production capability across partners, exploring additional revenue streams to support the ongoing costs of retaining capacity, fostering R&D collaborations to drive the development of mRNA products for inter-pandemic use, and recognizing the program’s potential to generate significant vaccine production during pandemics. Pandemic modeling projections for the high-output scenario indicated the potential for the program as a whole to deliver 600 million to 1.2 billion doses within a 10-year timeframe.

### 3.3. Session 3: Building the Ecosystem for the Development of New mRNA-Based Products in LMICs

The importance of mapping barriers and opportunities in the R&D of mRNA products, emphasizing their significant public health potential due to their versatility in targeting various diseases, was also discussed. Sustainable investment in both pandemic and endemic disease targets was highlighted, along with the need to expand the risk window for financial investments to support nascent manufacturing capacities in LMICs.

Afrigen Biologics & Vaccines’s (Afrigen) representative discussed the challenges encountered in developing mRNA technology in LMICs, particularly in Africa. Key challenges included time constraints, rapid skill development, equipment access, and facility establishment. Despite budget constraints, they successfully secured investments through strategic equity partnerships, emphasizing the importance of collective effort and collaboration among partners to ensure the sustainability of mRNA technology for public health purposes.

The Afrigen panelist underscored the importance of derisking mRNA capabilities and securing investments from strategic shareholders who understand the company’s mission and its potential impact on economic development. They aim for sustainable growth through a combination of shareholder investment and non-dilutive funding, highlighting the value of mRNA technology in both public health and commercial sustainability.

Institute Pasteur Tunis (IPT) shared their experience in establishing an mRNA tech transfer initiative focused on capacity building and developing mRNA vaccines and therapeutics for prevalent diseases in LMICs. Collaborations with organizations like Moderna and the Bill & Melinda Gates Foundation have been pivotal in advancing their research. Additionally, IPT highlighted a European Commission-funded project aimed at training the workforce in mRNA technology, encompassing the entire process from research to market access.

The Islamic Development Bank (IsDB) emphasized the need for coherence between private and public sector lending, noting that mRNA technology transfer requires a long-term commitment due to the complex vaccine development process. They stressed the importance of comprehensive financing packages involving multiple stakeholders and highlighted the need for coordinated planning and support to establish a regional vaccine supply and introduce new vaccines.

Instituto de Tecnologia em Imunobiológicos (Bio-Manguinhos), a public manufacturer, discussed the role of public–private partnerships in developing new products. They highlighted their tradition of collaborating with both large pharmaceutical companies and smaller firms, emphasizing the importance of pooling knowledge and resources. Despite funding restrictions, Bio-Manguinhos explores various alternative funding sources and has benefited from Brazil’s programs designed to foster public–private partnerships.

The IFC provided insights on supporting R&D investments and the paradigm shift brought about by platforms like mRNA technology. They emphasized the need for diverse models tailored to individual circumstances and highlighted the critical role of governments and venture capital in prioritizing and funding R&D. They stressed the importance of comprehensive business plans and collaboration among stakeholders to effectively advance R&D initiatives in developing regions.

Key findings from the session included the critical need for sustainable investment, the importance of derisking strategies, the role of strategic partnerships, and the necessity for diverse financing models tailored to the specific needs of LMICs. The discussions underscored that building a robust ecosystem for mRNA technology development requires a multifaceted approach involving public–private partnerships, innovative financing, and strong stakeholder collaboration.

### 3.4. Session 4: Addressing the Needs of a Nascent mRNA Industry in LMICs

Another discussion focus was on the challenges and opportunities in establishing a new mRNA vaccine manufacturing industry in LMICs, with panelists sharing insights on policy enablers and lessons learned from their ongoing partnerships. A recurring theme was the crucial role of governments in supporting the establishment of a new ecosystem, emphasizing the need for substantial financial investments and comprehensive support systems.

Kenya BioVax Institute highlighted the importance of governmental and funding institute recognition of the unique challenges faced by new biomanufacturers. These include the need for significant financial investment before marketable products are available, which often conflicts with the immediate needs of governments. They stressed that financing mechanisms should adopt an ecosystem approach, encompassing workforce recruitment, capacity development, regulatory process definition, and system strengthening to establish a nascent industry.

Sinergium Biotech, from Argentina, underlined the role of enabling policies and public–private partnerships in securing non-governmental investments. Long-term procurement commitments from the Argentinian government were crucial in shaping demand and ensuring alignment with national and regional strategies. These commitments mitigated risks and assured financing for infrastructure development, facilitating economies of scale and reducing the cost of goods.

The Biovac Institute, based in South Africa, reiterated the centrality of an ecosystem approach to manufacturing. They emphasized that government procurement commitments support manufacturer growth but longer-term commitments (e.g., 10-year vs. 2-year) provide better financial leverage. They also highlighted the necessity of a whole-of-government approach to policy changes, ensuring all governmental arms are informed to allow local manufacturers to compete effectively for tenders.

Despite government support, local demand may be insufficient for manufacturers in small countries. Torlak stressed the need for favorable policies, such as trade agreements, to support regional procurement. Regional demand can reduce fixed costs and ensure sustainability, which is crucial for new market entrants.

A key lesson from the COVID-19 pandemic is the importance of health security and the role of technology transfer initiatives in contributing to the manufacturing ecosystem. However, technology transfer initiatives often focus on local markets without incentivizing regional supply. New initiatives like the African Vaccine Manufacturing Accelerator (AVMA) aim to create incentives for partnerships with local manufacturers, emphasizing the procurement of locally/regionally produced products for both regional and global markets. The PAHO Revolving Fund for Access to Vaccines [12] has more than 40 years of experience in the Americas and is adapting its operations to encourage regional innovation and production projects and improve equitable access.

The IFC discussed financing instruments for the nascent industry, stressing a multifaceted approach based on the manufacturer’s maturity level. Considerations include product scalability, market access, and the type of funding needed (e.g., angel funding vs. equity investment). The IFC emphasized that strategic partnerships are crucial for manufacturing capabilities and regulatory pathways. Aggregating markets was identified as a key strategy for mitigating financial risks and enhancing returns on investment, highlighting the importance of regionalized manufacturing capacity to supply regional markets.

Key findings presented by the speakers included the necessity for sustainable investment, the importance of derisking strategies, the role of strategic partnerships, and the adoption of diverse financing models tailored to the specific needs of LMICs (Figure 4). The discussions highlighted the need for a comprehensive approach involving public–private partnerships, innovative financing, and strong stakeholder collaboration to build a strong mRNA technology development ecosystem.

### 3.5. Session 5: Envisaging New Financing Models for LMIC-Based Manufacturers

The IFC started this panel discussion by asking manufacturers to reflect on how to overcome funding barriers. They highlighted an ecosystem approach, the importance of consortia, and collaborations. Manufacturers reflected on the need for strategies for raising funds, commercializing products, and building R&D and manufacturing partnerships. The Biovac Institute stressed the necessity of targeting the right investors and using appropriate language better suited to facilitate such technical discussion between the scientific and economic audiences. They suggested a blended financing approach, combining grant funding with MDB/development finance institution (DFI) financing, to support R&D, biomanufacturing development, and product commercialization, facilitating comprehensive end-to-end funding.

During the COVID-19 pandemic, philanthropic funding played a crucial role in supporting R&D and manufacturing. A similar mechanism to the pandemic fund, designed to strengthen LMIC capacities for pandemic prevention and response, could be developed. It was noted that “fragmentation creates opportunities for funders to have more supplier power”. A neutral global agency could enhance biomanufacturers’ procurement power by aggregating and coordinating initiatives, thereby increasing credibility and funding leverage. MDBs could develop partnerships and mechanisms to attract a pooled fund that biomanufacturers can access, supporting financing needs.

Further discussion highlighted the importance of coordination among various capital pools provided by MDBs, DFIs, and governmental institutions. Aggregation should include support from funders to ensure organization and enable widespread assistance. Exclusion lists for certain funders could impede progress, underscoring the need for neutral partners to facilitate support for all stakeholders. The debate emphasized that innovative financing models and collaborative efforts are essential for supporting the growth and sustainability of LMIC-based biomanufacturers.

## 4. Conclusions and Future Perspectives

The convening of this meeting marked an inaugural opportunity to engage in discussions regarding sustainable end-to-end financing along the value chain of mRNA-based products. Rather than focusing solely on product transfer, the emphasis lies on platform technology transfer, heralding a new model for advancing mRNA technology that is rooted in a multilateral model for product development and sustainable production.

Developing vaccines presents significant financial challenges, particularly for those aimed at complex diseases, such as emerging infectious diseases. These vaccines typically require lengthy, complicated, and expensive clinical trials [13]. Furthermore, the financial returns on vaccine investments can be unpredictable due to various factors, including development issues and market dynamics. The disparity between the high costs of development and the limited revenue potential in certain markets—along with the uncertainty of scenarios like pandemics—can discourage investment in vaccine research, especially for diseases that disproportionately affect vulnerable populations [14,15].

To address the risks and uncertainties associated with such investments, governments and international organizations have established various financial mechanisms and incentives [16,17]. The importance of these financial tools became especially evident during the COVID-19 pandemic, highlighting the need for effective support in vaccine development [18].

In this context, the meeting focused on discussing the necessity of financial mechanisms and incentives that target the research and development of mRNA vaccines for PPPR, aligned with the objectives of the mRNA TT Programme and the capabilities of its partners [19,20,21]. The meeting sessions highlighted the increasing significance of financial models that cater to both short-term and long-term needs in mRNA vaccine R&D. Panelists consistently pointed out that achieving financial sustainability for mRNA production in LMICs necessitates a blend of public and private investment, supported by strong government backing. Blended financing—a combination of grants, concessional loans, and equity investments—was emphasized as a crucial strategy for promoting innovation while managing risks. MDBs indicated that implementing such models could greatly lower financial obstacles and stimulate investment in emerging vaccine industries, especially during non-pandemic times. MDBs would also focus on adjusting their policies and tools to support a rapid and equitable pandemic response.

In line with ongoing pandemic preparedness efforts [17,18,19,21,22], like the WHO/MPP mRNA TT Programme, the discussions underscored that scalable mRNA production in LMICs will require more than just financial input. Establishing a supportive ecosystem—including solid regulatory frameworks, scientific initiatives, workforce development, and regional procurement systems—was deemed vital for maintaining vaccine manufacturing capabilities. The need to diversify revenue streams to support production and R&D activities between pandemics was also highlighted, with panelists advocating for long-term partnerships to enhance resilience and lessen reliance on single-source funding.

The early stages of mRNA technology were underscored by the groundbreaking potential demonstrated by the COVID-19 mRNA vaccine, igniting efforts to democratize access to mRNA technology through collaboration and network building. Despite the challenges, manufacturing panelists expressed optimism. They emphasized the major benefits of being part of a multilateral TT program: collaboration, coordination, and the consortium approach applied to the mRNA program, which supports the sharing of ideas and knowledge that contributes to increased yield and lowered production costs. Any breakthrough and improvements by one manufacturer can benefit all and lower production costs. The 15 program partners in strategic regional entities further reduce competition and also allow for regional supplies of products when they become available.

The discussions highlighted that to achieve fair access to vaccines, it is essential to establish regional manufacturing hubs that can cater to both local and global needs. Creating these hubs would not only bolster global health security but also stimulate economic growth in LMICs. Recommendations stressed the importance of collaboration among MDBs, governments, and private investors to enhance manufacturing capacity while ensuring vaccines are distributed equitably.

By nurturing innovation-driven ecosystems and supporting local manufacturers, stakeholders can help build a more resilient global health system, prepared to address future pandemics and health crises effectively. These future considerations emphasize the urgent need for ongoing dialog, innovation, and investment in mRNA vaccine technology. The conclusions drawn from the meeting will lay the groundwork for continued collaboration among key stakeholders, steering future initiatives toward a more equitable, sustainable, and secure global health system.

An end-to-end approach is advocated to encompass the entirety of the mRNA production process beyond mere manufacturing. Strengthening systems beyond manufacturing is deemed essential, encompassing the development of human resources and regulatory capabilities. Regional and global collaboration is identified as a catalyst for accelerating vaccine development, creating value beyond the sum of individual entities involved.

Lastly, it is underscored that sustainability hinges on positioning mRNA partners in regional and global markets, necessitating the involvement of various stakeholders, including governments, funders, venture capitalists, and philanthropists. This meeting marks the initiation of a concerted effort towards sustainable financing in the mRNA program, with a commitment to addressing the needs and perspectives of all stakeholders involved.

Amidst technological advancements, there is a pressing need to develop innovative financing models to support these endeavors. Notably, this meeting represents the first gathering of all multilateral development banks to deliberate on bolstering mRNA production capacity, setting the stage for an essential dialog among a diverse array of partners. As the technological landscape evolves, it is evident that our thinking must adapt accordingly.

Key themes emerging from this dialog include the imperative to rethink financing tools, with multilateral development banks prepared to adjust their strategies to accommodate the public good component inherent in mRNA production. Moreover, there is a recognition of the pivotal role of the public sector in facilitating the flourishing of mRNA technology, necessitating a better understanding of public–private linkages. Furthermore, the value of local manufacturing must be factored into funding evaluations, considering its dual role in addressing local health needs and responding to pandemics, and support should encompass a whole-of-government approach, including industry, infrastructure, regulations, and health policies.

## Figures and Tables

**Figure 1 vaccines-13-00112-f001:**
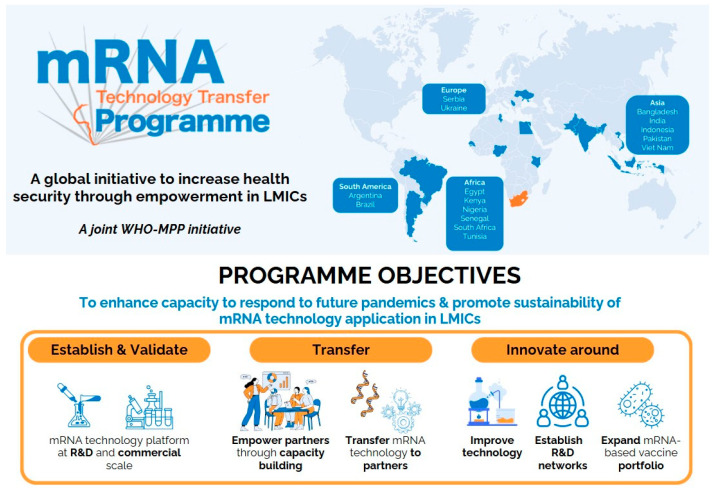
mRNA Technology Transfer Programme objectives.

**Figure 2 vaccines-13-00112-f002:**
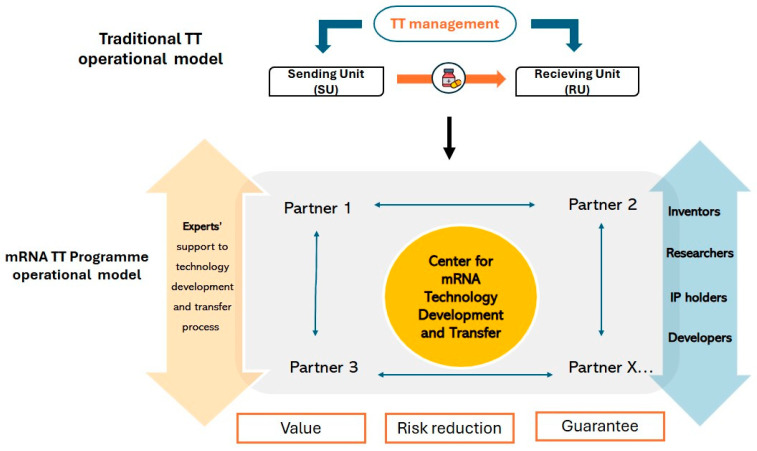
mRNA Technology Transfer Programme operational model. TT: technology transfer.

**Figure 3 vaccines-13-00112-f003:**
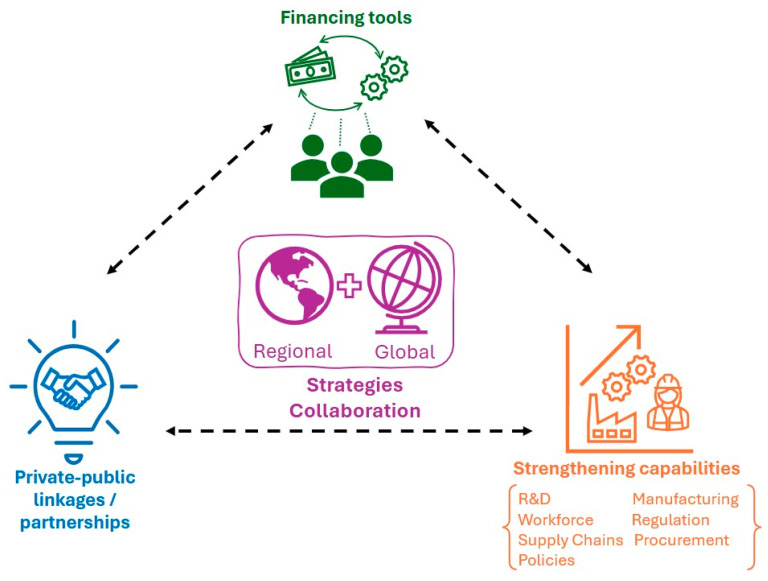
Key themes highlighted as enablers for sustainable end-to-end biomanufacturing.

**Figure 4 vaccines-13-00112-f004:**
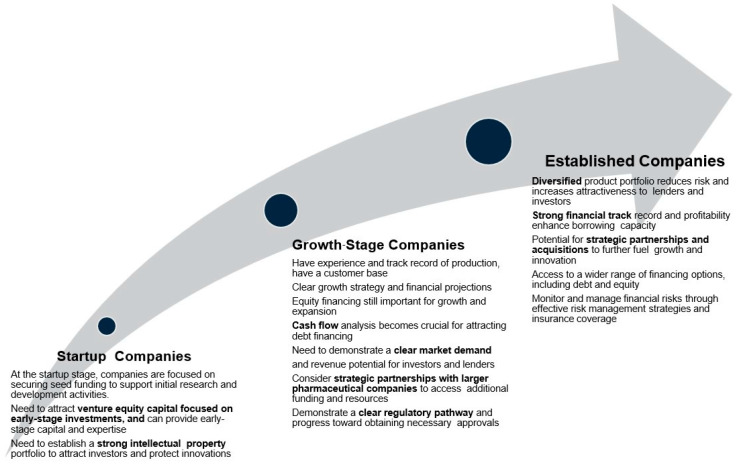
Considerations of biomanufacturers’ maturity, financing mechanisms, and opportunities to support financing manufacturers.

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
