# Peer review of "Investment Opportunities for mRNA Technology in Low- and Middle-Income Countries: Key Findings and Future Perspectives"

_vaccines, 2025, doi:10.3390/vaccines13020112_

Round 1
Reviewer 1 Report
Comments and Suggestions for Authors
This manuscript presents a stenograph and selection of key points discussed in April 2024 at a hybrid meeting organized by WHO, PAHO and MPP during the World Bank Spring Meetings on financing mRNA-based technologies in low- and middle-income countries (LMICs).
Although this manuscript is an important call for cooperation and adjustment of the economic models in order to improve the level of health care in LMICs, it does not meet the requirements of Vaccines MDPI for publication, namely, the manuscript does not meet the requirements of either «an article» or «a review» type. In this regard, the reviewer does not recommend its publication in Vaccines MDPI.
Below are the typos identified by the reviewer:
Line 38: Despite the progress in expanding routine immunization programs, the majority of LMICs, and vaccines supplied in the region are predominantly produced in a few countries
Please, rephrase the sentence, the semantic role of «the majority of LMICs » in this sentence is not clear.
Line 67: (7) should be [7]
Line 162: «Figure 2. Key themes highlighted as enablers for sustainable end-to-end biomanufacturing. .»
Please, delete the second dot
Line 289: Please clarify what Torlak is.
Author Response
Thank you for your comments. We much appreciate your suggestions to increase the quality of this work. We have updated the manuscript to Conference Report type as per Vaccines guidelines.
We have modified the text in lines 39-45 for clarity: “However, progress in vaccination coverage varies significantly across countries, especially in low- and middle-income countries (LMICs), where vaccination rates are often hindered by vaccine inequities driven by affordability, availability and weakened health systems”.
We have also corrected the reference format, which is now in line 92 as reference 9, as well as removed the additional dot.
We have created supplementary table 1, mentioned in line 132 to clarify the list of institutions that took part in the meeting and their type, which includes Torlak.
Reviewer 2 Report
Comments and Suggestions for Authors
I admit that this article is an interesting survey, but Vaccines Research Journal recommends that it be written in an academic research format.
Introduction:
1. Please provide several gaps in past research and explain how this study fills these gaps.
2. Please clearly indicate the incremental contribution of this article.
3. Please indicate your research question
Literature discussion:
1. Please add the literature discussion paragraph, which is missing in this survey.
2. Please discuss the relevant literature of this article and propose possible answers to the research questions.
Research methods:
1. Please indicate the research method used in this investigation.
Conclusion:
Please add incremental contributions to this article rather than describing what is already known
Comments on the Quality of English LanguageThe English could be improved to more clearly express the research.
Author Response
Thank you for your comments. We much appreciate your suggestions to increase the quality of this work.
We have updated the manuscript to Conference Report type as per Vaccines guidelines. The article objective is to describe the main discussions and findings from this meeting in relation to financial opportunities for mRNA vaccine R&D considering pandemic preparedness, prevention and response. No survey was conducted for that. We adjusted the objectives for clarity in lines 118-122, “This article aims to describe the meeting discussions related to challenges associated with financing mRNA vaccine manufacturing, the potential financial opportunities and mechanisms that can enable multilateral partnership for the development and manufacturing of innovative products and list the findings in relation to financial opportunities for mRNA vaccine R&D considering pandemic preparedness, prevention and response (PPPR).”
To make clear we added a section related to the meeting design (lines 124-153), which includes information on how the meeting was planned and how the main findings and discussions of each session will be described. We also included as supplementary material the list of participating institutions and their type and the meeting agenda.
We also switch the conclusion section to conclusion and future recommendations to expand the contributions of the meeting findings. This is now presented in lines 398-471.
Reviewer 3 Report
Comments and Suggestions for Authors
This is a meeting report on a hybrid meeting held on April 2024 on the possibilities to finance mRNA-vaccine production investment in low and middle-income countries. The meeting report, even though quite extensive, is well-written. I have no doubts of the accuracy of the report. Evidently, the issues relating to the patent-rights of mRNA vaccines were not thoroughly discussed in the meeting.
Author Response
Thank you for your comments. We much appreciate your suggestions to increase the quality of this work. We have updated the manuscript to Conference Report type as per Vaccines guidelines to make it clear to the readers.
The meeting was focused only on the discussion of financial opportunities for mRNA vaccine R&D considering pandemic preparedness, prevention and response. Points related to investment and financial models for local manufacturing of mRNA products in LMIC, capital and operational expenditures and production capacity focused on PPPR, development ecosystem for mRNA-based products and needs for an mRNA industry in LMIC were the key topics of discussion in accordance with the meeting agenda now present as supplementary table 2. In that sense, patent related issues were not a proposed topic of discussion, and neither was it raised as an issue related to financial opportunities during the discussions. We adjusted the objectives for clarity in lines 118-122, “This article aims to describe the meeting discussions related to challenges associated with financing mRNA vaccine manufacturing, the potential financial opportunities and mechanisms that can enable multilateral partnership for the development and manufacturing of innovative products and list the findings in relation to financial opportunities for mRNA vaccine R&D considering pandemic preparedness, prevention and response (PPPR).”
To make clear we added a section related to the meeting design (lines 124-153), which includes information on how the meeting was planned and how the main findings and discussions of each session will be described. We also included as supplementary material the list of participating institutions and their type and the meeting agenda.
Reviewer 4 Report
Comments and Suggestions for Authors
The study contains valuable information but lacks a clear structure, making it difficult to follow the flow of arguments. Consider organizing the content into distinct sections, such as "Introduction," "Methods," "Findings," "Discussion," and "Conclusion," for improved readability.
Few comments
1)The context and purpose of the study are unclear. Include a brief introduction at the beginning that explains the goals of the session, the significance of mRNA technology in LMICs, and how this ties into the broader public health landscape.
2)terms such as "mRNA TT Programme," "CapEx," and "OpEx" should be briefly defined or explained upon their first mention to ensure accessibility for a wider audience.
3)The authros references various stakeholders (e.g., Afrigen, IPT, AIIB, IFC) but does not delve deeply into their specific contributions or challenges. Providing more detail on their roles and the outcomes of their involvement would make the analysis more robust
4)While the authros emphasizes pandemic preparedness, it does not address how these investments align with broader global health goals, such as achieving universal health coverage. Including this perspective would add depth.
Author Response
Thank you for your comments. We much appreciate your suggestions to increase the quality of this work. We have updated the manuscript to Conference Report type as per Vaccines guidelines to make it clear to the readers.
The article objective is to describe the main discussions and findings from this meeting in relation to financial opportunities for mRNA vaccine R&D considering pandemic preparedness, prevention and response. We adjusted the objectives for clarity in lines 118-122, “This article aims to describe the meeting discussions related to challenges associated with financing mRNA vaccine manufacturing, the potential financial opportunities and mechanisms that can enable multilateral partnership for the development and manufacturing of innovative products and list the findings in relation to financial opportunities for mRNA vaccine R&D considering pandemic preparedness, prevention and response (PPPR).”
To make clear we added a section related to the meeting design (lines 124-153), which includes information on how the meeting was planned and how the main findings and discussions of each session will be described. We also included as supplementary material the list of participating institutions and their type and the meeting agenda.
Lines 218-224 were updated to provide more clarity regarding the definition of Capex and Opex. For mRNA TT Programme we have included Figure 1 (line 71) which states the Programme focus and objectives. In addition, references 7 and 8, already present in the text refer to further explanation of the programme.
We also switch the conclusion section to conclusion and future recommendations to expand the contributions of the meeting findings. This is now presented in lines 398-471.
Round 2
Reviewer 1 Report
Comments and Suggestions for Authors
As a conference report, this manuscript is suitable for publication in Vaccines MDPI. The authors corrected typos
Author Response
We appreciate the reviewer’s suggestions to enhance our manuscript's quality. We are glad the changes meet the expectations.
Reviewer 2 Report
Comments and Suggestions for Authors
Thanks to the author for revising the article based on my previous comments. The quality of the article now looks good. This discussion section highlights the unique features of the article; however, it lacks any substantive analysis or critical discussion. As a result, it does not demonstrate how this work differs from existing literature. I recommend the author incorporate a more in-depth discussion that contrasts their findings with previous studies and identifies the gaps or contributions this research makes to the field.
Author Response
We thank the reviewer for the suggestions and comments. The manuscript describes the discussions held during the meeting regarding the challenges of financing mRNA vaccine manufacturing. It also explores the perspectives of the participants on potential financial opportunities and mechanisms that could facilitate multilateral partnerships for developing and producing innovative products. Additionally, the manuscript lists findings related to financial opportunities for mRNA vaccine research and development, particularly in the context of pandemic preparedness, prevention, and response.
To emphasize the significance of the findings and discussions from the meeting, we included lines 405-422 to provide context for why the meeting was organized, supported by relevant literature references. We hope this will clarify the existing gaps and highlight the contributions of the meeting's findings.